# Simulation of the COVID-19 epidemic on the social network of Slovenia: Estimating the intrinsic forecast uncertainty

Žiga Zaplotnik[1]*, Aleksandar Gavrić[2], Luka Medic[1]

**1** Faculty of Mathematics and Physics, University of Ljubljana, Ljubljana, Slovenia, **2** Department of Gastroenterology and Hepatology, University Medical Center Ljubljana, Ljubljana, Slovenia

* ziga.zaplotnik@fmf.uni-lj.si

## Abstract

In the article a virus transmission model is constructed on a simplified social network. The social network consists of more than 2 million nodes, each representing an inhabitant of Slovenia. The nodes are organised and interconnected according to the real household and elderly-care center distribution, while their connections outside these clusters are semi-randomly distributed and undirected. The virus spread model is coupled to the disease progression model. The ensemble approach with the perturbed transmission and disease parameters is used to quantify the ensemble spread, a proxy for the forecast uncertainty.

The presented ongoing forecasts of COVID-19 epidemic in Slovenia are compared with the collected Slovenian data. Results show that at the end of the first epidemic wave, the infection was twice more likely to transmit within households/elderly care centers than outside them. We use an ensemble of simulations ($N = 1000$) and data assimilation approach to estimate the COVID-19 forecast uncertainty and to inversely obtain posterior distributions of model parameters. We found that in the uncontrolled epidemic, the intrinsic uncertainty mostly originates from the uncertainty of the virus biology, i.e. its reproduction number. In the controlled epidemic with low ratio of infected population, the randomness of the social network becomes the major source of forecast uncertainty, particularly for the short-range forecasts. Virus transmission models with accurate social network models are thus essential for improving epidemics forecasting.

## Introduction

The ongoing COVID-19 epidemic has revealed a major gap in our ability to forecast the evolution of the epidemic. The most common approach to simulate the epidemic dynamics is using compartmental models of susceptible (S), infectious (I) and recovered (R) population, i.e. SIR models [1, 2]. These are described by a system of differential equations given some predefined parameters, such as probability of the disease transmission and the rate of recovery or mortality. Another variation of the SIR model, which is more applicable to some viral diseases, is a SEIR model, which accounts also for the exposed (E) population, representing infected but less

**Funding:** Ž.Z. was funded by the Slovenian Research Agency, project J1-9431. Ž.Z. is

supported also by ARRS Programme P1-0188.
http://www.arrs.si/sl/ The funders had no role in
study design, data collection and analysis, decision
to publish, or preparation of the manuscript.

**Competing interests:** The authors have declared
that no competing interests exist.

infectious subjects or subjects not infectious at all [3]. The SEIR model is often combined with
activation functions to smoothly model social factors affecting virus spread and the disease
progression.

A major setback of the deterministic epidemic models is that they are only suitable for suffi-
ciently large populations with large number of infectious subjects, in which case the assump-
tion of random mixing and homogeneous spread of the virus is valid [4]. However, for
coronaviruses including SARS-CoV-2, there is evidence that some infectious cases, the so
called superspreaders, spread virus more than others [5, 6]. Their role is of the utmost impor-
tance when the population of infectious is small, i.e. in the initial uncontrolled phase of an epi-
demic and in its final controlled phase. In these cases, the deterministic SEIR models are
unable to properly describe the intrinsic uncertainty of the epidemics forecast related to het-
erogeneous connectivity of the social network and to heterogeneous disease progress of the
infected population.

In order to account for the superspreading nature of the new coronavirus (SARS-CoV-2)
and to properly estimate the forecast uncertainty, we use network-based approach to simulate
the virus spread. The simplified social network consists of more than 2 million nodes with a
total of up to 20 million undirected connections, representing the population of Slovenia and
the contacts of its inhabitants, with realistic distinction between household and outer contacts.
Despite being computationally more expensive, the advantage of the network approach is that
it allows direct simulation of intervention measures, contact-tracing strategies and other strat-
egies of the the virus containment [7, 8] as well as the lockdown-exit strategies.

The network epidemiology research has heavily advanced in the last three decades [8]. A
variety of different network types has been developed [9], however the breakthrough of social-
network approaches has been halted by the insufficient social data and epidemiological data
which would allow to verify different assumptions in the generation of social networks [10].
An exception to this includes studies, where the social network was generated based on the
comprehensive contact survey data [11, 12]. Nevertheless, network models have often been
criticised for the large number of parameters they require [13].

In this study, we perform an ensemble-of-simulations of the virus spread over the social
network. Since the network is randomly generated in each simulation, the evolution of the epi-
demics will differ between simulations. Different nodes are infected at initial time, while each
simulation also uses different virus transmission parameters and disease progress parameters,
which are perturbed according to their known distributions. This approach allows to estimate
the uncertainty of the epidemic forecasts in the case of controlled epidemic and uncontrolled
epidemic. To our knowledge, no study has ever compared the impact of network perturba-
tions, transmission parameters perturbations and other perturbations on the uncertainty of
the epidemic forecast.

The article is organised as follows. Methodology section describes the social network
model, the virus transmission model and the coupled disease progression model. The probabi-
listic ensemble forecast of the COVID-19 epidemic for Slovenia and the contribution of differ-
ent model components to the total forecast uncertainty are described in section Results,
followed by the Discussion, conclusions and further outlook.

## Methodology

### Social network model

The social network model of the population of Slovenia distinguishes household connections
and connections outside households. A total of $N = 2045795$ nodes is used in the social net-
work. The number of $k$-person households is given in Table 1 and is based on the data of

**Table 1. Households size distribution in Slovenia.**

| k persons in household | number of k-person households |
| --- | --- |
| 1 | 269898 |
| 2 | 209573 |
| 3 | 152959 |
| 4 | 122195 |
| 5 | 43327 |
| 6 | 17398 |
| 7 | 6073 |
| 8 | 3195 |

Statistical Office of Republic of Slovenia [14]. There are approximately 100 elderly care centers in Slovenia with a total of approximately 20000 residents. Each elderly care center is assumed to include 8 distinct groups of 25 people. Average household/care group consists of 2.5 people in Slovenia so the average number of contacts per person within household is 1.5.

In normal conditions, contact number distribution follows power law with fat tails [15], which are associated with superspreader events, e.g. large public gatherings such as sport and cultural events. However, since all public events are canceled in the event of the COVID-19 epidemic, these fat tails are cut off [16] and the topology of the social network changes substantially. In conditions without large public gatherings, it is reasonable to assume that certain people still have much larger number of contacts than others. The studies of social mixing, e.g. POLYMOD study of social interactions within 8 European countries, typically report negative binomial distribution of the number of contacts [17, 18]. We assumed mean number of contacts outside households to be 13.5 with standard deviation of 10.5. Instead of negative binomial distribution, we rather use smooth gamma distribution, which resembles the shape of the binomial distribution but has some useful mathematical properties, which will be exploited in the continuation. Thus, we model the connectivity, i.e. the number of outer contacts per person, using the gamma probability distribution, which is essentially an exponential distribution

$$p(x; k, \theta) = \frac{1}{\Gamma(k)\theta^k} x^{k-1} e^{-\frac{x}{\theta}}. \tag{1}$$

In this study, we use Gamma distribution with shape parameter $k = 1.65$ and scale parameter $\theta = 4.08$ for the initial setup in order to mimic the above-mentioned negative binomial distribution. This gives an average number of 13.5 outer contacts per person per day (Fig 1). Together with 1.5 family contacts per person per day, the total number of contacts per person per day is 15. Here, we assume that the average number of contacts is the same for each age group, despite studies showing that elderly have reduced number of contacts [19]. The average contact number per person per day varies for different countries, however a total of 15 contacts per day is a reasonable assumption for Slovenia based on the numbers reported for other Central European countries [18] and based on other contact surveys [11]. We also assume quasi-static social network, i.e. only 20% of contacts are new every day, and the remaining 80% are static. This choice is a first guess, justified by the fact that only around 20% of all daily contacts last less than 15 minutes [18]. These can be regarded as random sporadic contacts. Distancing measures to mitigate COVID-19 can be imposed by decreasing parameter $\theta$, which also decreases the average number of outer contacts (Fig 1).

Fig 2 shows an example of the connectivity change of a minimised network with 88 nodes clustered on a circle with the real household distribution taken into account.

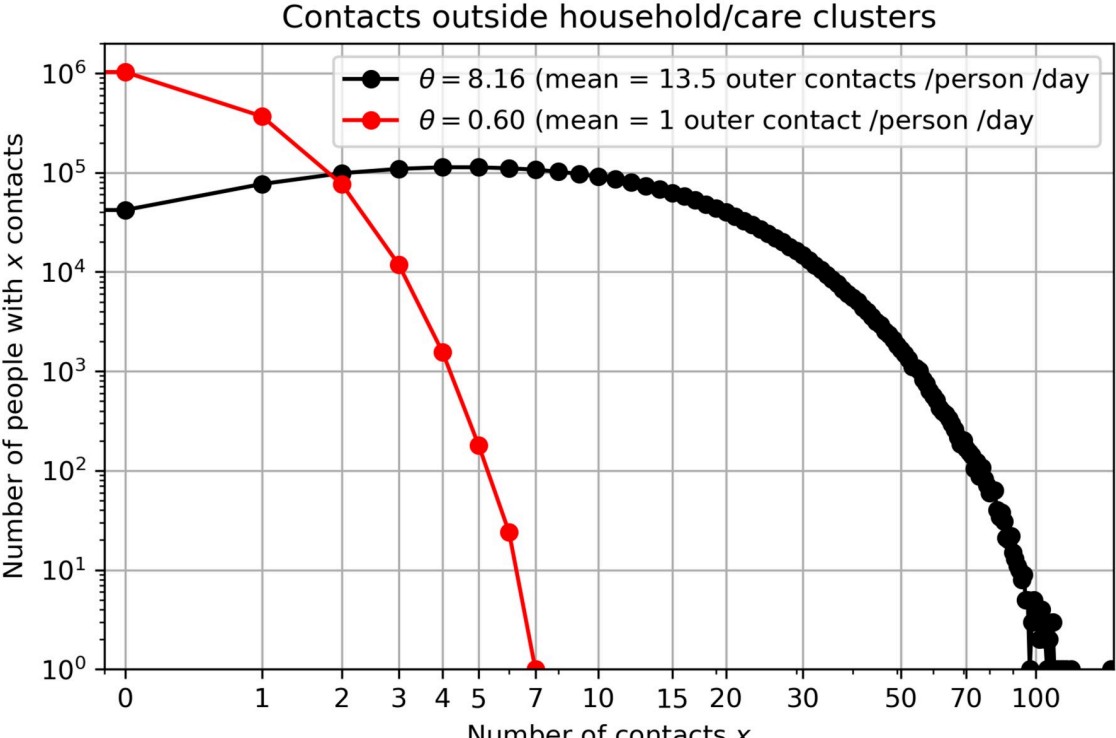

**Fig 1. Distribution by the number of contacts.** Number distribution $N(x) = p(x)N$ by the number of contacts in the social network model. The black graph shows the assumed distribution of people with a given number of outer contacts in a normal, non-epidemic phase, while the red graph presents reduced number of outer contacts in the case of a social distancing measures.

Technically, we connect the graph in the following way:

1. number of outer contacts for each node is randomly drawn from Gamma distribution (Eq 1). If node $i$ has $x_i = 0.33$ contacts per day, it means that it will have 0 contacts 2/3 of the time and 1 contact 1/3 of the time of the simulation;

2. for each node $i$, we randomly assign the connections to $x_i$ other nodes, where $x_i$ is the number of contacts of node $i$. However, not every node has the same probability of being picked as a neighbour. Node $j$, which has $x_j$ contacts, is picked as a neighbour with probability $x_j N(x_j)/T$, where $N(x_j)$ is the number of nodes with $x_j$ contacts and $T$ is the total number of contacts in the network ($T$ is twofold the number of connections). Sampling over Gamma distribution (Eq 1) gives us a distribution of $N(x) = p(x)N$. When picking the neighbours, we actually sample the same Gamma distribution times $x$, i.e.

$$p_n(x) = p(x; k, \theta)x = \frac{1}{\Gamma(k)\theta^k} x^k e^{-\frac{x}{\theta}} \propto p(x; k+1, \theta). \tag{2}$$

3. The shape of the social network is changing at every timestep of the simulation (80% of connections static, 20% changing) to account for random sporadic contacts. (a) The number of contacts of node $i$ is fixed (randomly jumps between $\lfloor x_i \rfloor$ and $\lceil x_i \rceil$ based on the value of $x_i$). For example, if a node has 0.33 contacts per day, 1 contact is picked with probability 1/3 and 0 contacts with probability 2/3. (b) The social network is partially rewired at every time step to account for superspreaders mobility.

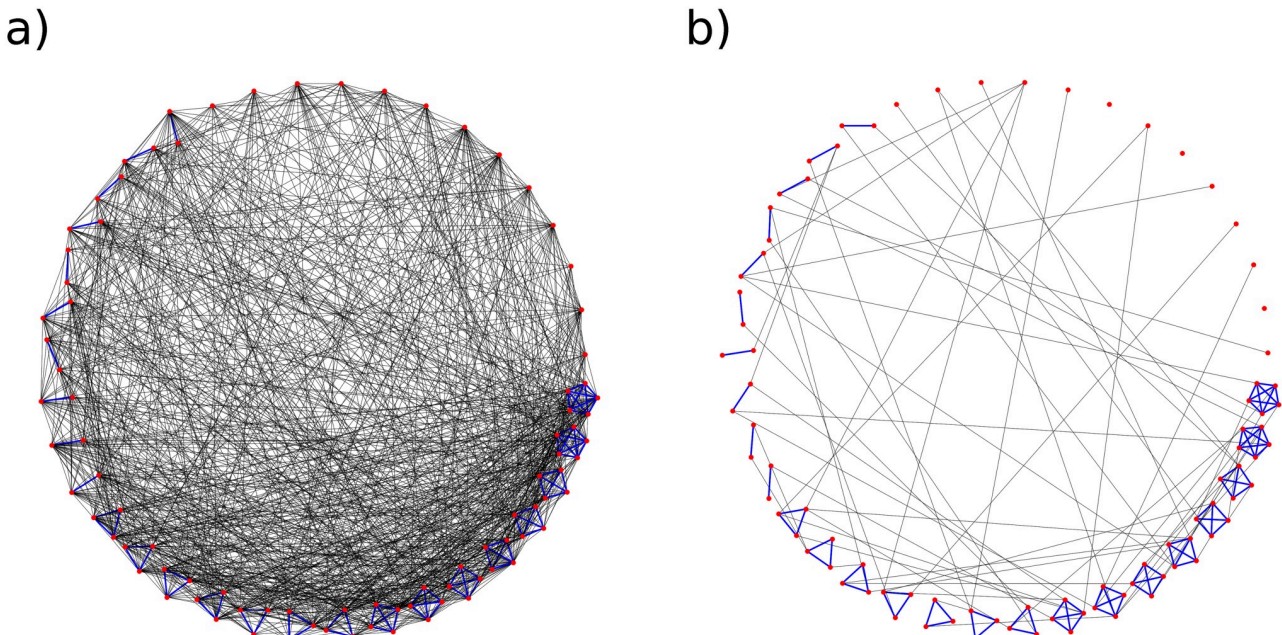

**Fig 2. Social network connectivity.** Connectivity of the social network of $N = 88$ nodes for a) densely connected graph, where each node has on average 15 contacts per day (1.5 family and 13.5 outer contacts, $\theta = 22.5$) and b) sparsely connected graph, where each node has on average 2.5 contacts per day (1.5 family and 1 outer contact). Red dots are nodes, blue lines represents household connections and black lines outer connections. The graph represents a minimized version of the social network used in the virus spread simulation.

An important advantage of our network approach is that the nodes are not connected randomly through "half-links" (directed connections linking egos to their contacts, the alters), such as in the vast majority of modelling studies, where the who-acquires-infection-from-whom matrices were constructed based on the egocentric data [9, 11]. Instead, nodes are fully-linked.

**Compartments.** Similarly as in the deterministic SEIR model, we divided the population into following compartments: susceptible, infected (exposed), infectious and recovered. The latter are assumed to be immune at least for the time period of the simulation. In the network model, a susceptible node becomes exposed (infected) with a certain probability (called an attack rate) when it is in contact with an infectious node. After a certain period of time (defined in the continuation), the infected node progresses into infectious state. In the accordance with the chosen compartmental division, the state of each mode is updated at every time step.

### Virus transmission model

**Reproduction number $R_0$.** The basic reproduction number $R_0$ provides information on the average speed of virus transmission in an uncontrolled phase of the epidemic. Different methodologies produced different results, however the majority of reported $R_0$ for SARS-CoV-2 is within 2 and 4. Here, we use median reported $R_0$ from a number of studies, as well as its median confidence intervals, i.e. $R_0 = 2.68$ (95% confidence interval (CI) 2-3.9). This approach is not the optimal one, since we are trading accuracy for precision. The published $R_0$ values as well as our deduced $R_0$ distribution is shown in Fig 3a and 3b. The optimal log-normal distribution should thus match the following conditions: $\text{CDF}(R_0^L; \mu, \sigma, \Delta x) = 0.025$, $\text{CDF}(R_0^U; \mu, \sigma, \Delta x) = 0.975$, and $\text{median}(\text{CDF}) = \exp(\mu) = R_0$, where $R_0^L$ and $R_0^H$ are lower and

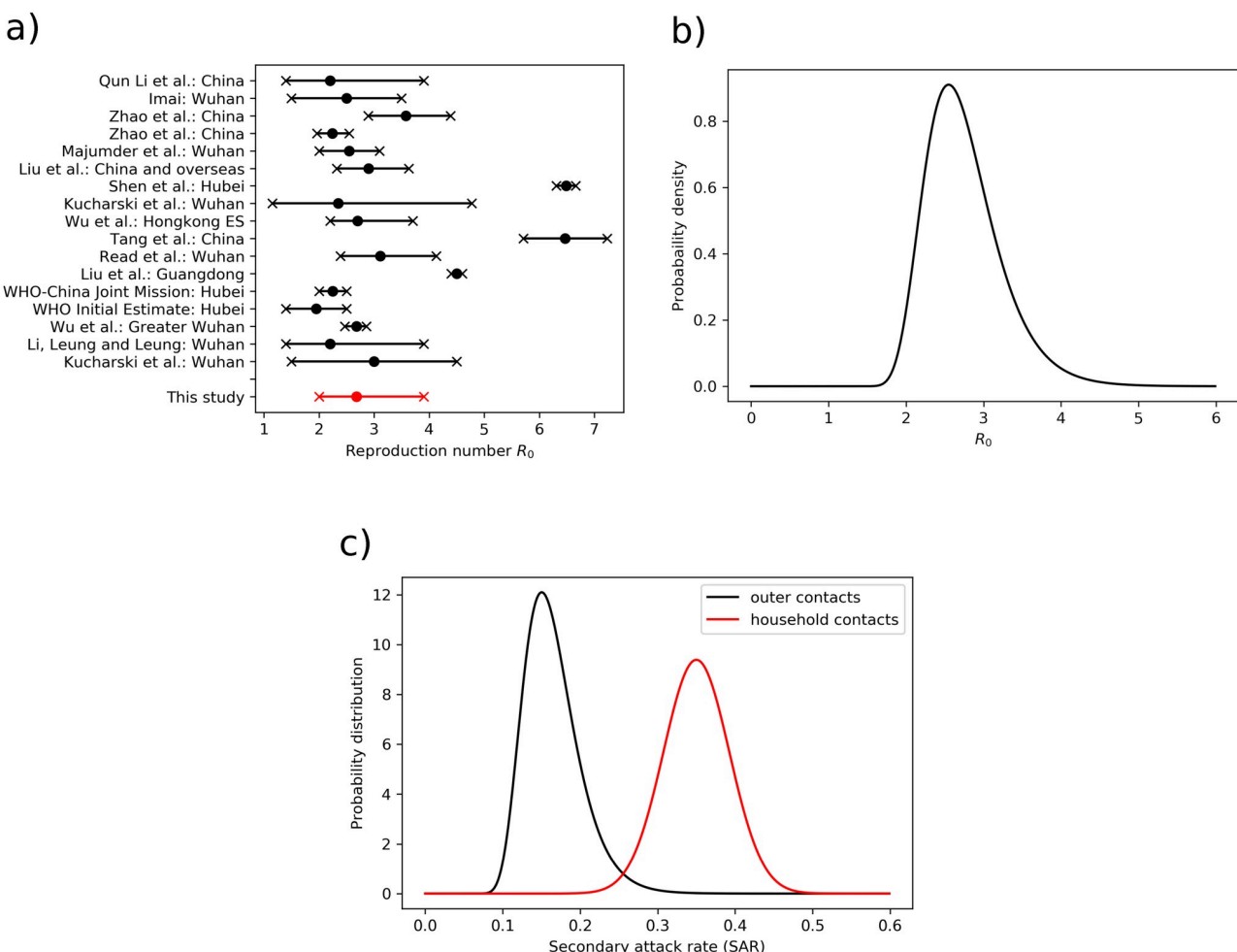

**Fig 3. Basic reproduction number and secondary attack rate.** a) Basic reproduction number $R_0$ (median and 95% confidence interval), reported in a number of studies for different locations [20–30] and references therein. b) Log-normal probability density function of the basic reproduction number, used for ensemble simulations. c) Probability distribution of secondary attack rates for household contacts and outer contacts.

upper boundaries of $R_0$, CDF stands for log-normal cumulative distribution function. Then, we define a quadratic cost function, which includes all the above criteria, and by minimizing it, we obtain the optimal parameters for log-normal distribution: $\Delta x = 0.36$ and $\sigma = 1.14$.

**Attack rate.** In general, the basic reproduction number $R_0$ can be decomposed into the secondary attack rate times the number of contacts. The secondary attack rate ($SAR$) is defined as the probability that an infection occurs among susceptible people within a specific group (i.e. household contacts or other contacts outside households). The measure can provide an indication of how social interactions relate to the transmission risk. We can further decompose the $R_0$ into the household risk of infection and outer risk of infection [31]

$$R_0 = SAR_h N_h + SAR_c N_c, \tag{3}$$

where $SAR_h$ and $SAR_c$ are secondary attack rates within household and outside household (outer contacts), respectively. $N_h$ and $N_c$ are the numbers of household contacs and outer contacts. Here, one must notice that the above estimation of $SAR_h$ assumed homogeneous mixing,

while the network model is heterogeneous with

$$R_0 = SAR_h \left( \langle N_h \rangle + \frac{\mathrm{Var}(N_h)}{\langle N_h \rangle} \right) + SAR_c \left( \langle N_c \rangle + \frac{\mathrm{Var}(N_c)}{\langle N_h \rangle} \right), \tag{4}$$

where $\langle \cdot \rangle$ denotes mean and $\mathrm{Var}(\cdot)$ denotes variance of the distribution of the number of contacts. To be consistent with [31], we stick with formulation (3).

The study of Liu et al. [31] suggests $SAR_h$ value of 35% (95% CI 27-44%) for SARS-CoV-2. The distribution of $R_0$ is given in the previous paragraph. It holds: $SAR_c = (R_0 - SAR_h N_h)/N_c$. This gives a transmission efficiency of $SAR_c = 16\%$ (95% CI 10.8-25.1%), in line with the recent estimate of the COVID-19 outbreak in an Israeli high school [32]. Fig 3c shows probability distributions of secondary attack rates as used in the ensemble of simulations.

If the social infectious period is $T_{inf} \approx 5$ days (check Infectious period), we can assume that the daily risk of getting infected from a certain household member is $SAR_{h,daily}$ where $1 - (1 - SAR_{h,daily})^{T_{inf}} = SAR_h$ and

$$SAR_{h,daily} = 1 - \exp \left( \frac{\ln (1 - SAR_h)}{T_{inf}} \right) \tag{5}$$

being equal 8.3% (95% CI 6.1-10.9%). Similarly, we compute $SAR_{c,daily} = 3.4\%$ (95% CI 2.3-5.6%).

Some studies have concentrated only on the symptomatic secondary attack rates and have shown relatively smaller numbers: 0.45% (95% CI 0.12%-1.6%) among all close contacts and 10.5% (95% CI 0.12%-1.6%) among household members [33]. However, these numbers cannot reproduce the reported $R_0$ between 2 and 3.9 with any realistic number of contacts. Another study shows similar attack rates to what we use here [34].

The attack rate affects the virus transmission as follows. At each timestep of the simulation (every 1 day), the susceptible contacts of each infectious individual are randomly infected with probability $SAR_{h,daily}$ or $SAR_{c,daily}$, depending whether the contact occurs within household or outside it.

## Disease progression model

A simplified sketch of the disease progression model is shown in Fig 4a. When a certain individual (node) gets infected, incubation period starts and several days will pass until the symptom onset (for symptomatic infection). The majority of the infected people recovers at home/elderly care centers, some cases with fatal outcome are only given palliative care, while certain individuals are admitted to hospital in the following days. Several outcomes are possible: recovery after normal hospitalisation, recovery after intensive care unit hospitalisation, and death. Note that for every node, the illness evolves differently (according to one of the above scenarios) and based on the probability distributions described in the following subsections.

**Case fatality ratio.** The baseline case fatality ratio (CFR), i.e. the fatality ratio among all positively tested, is assumed 1.38% (CI 1.23-1.53%) [35, 38], similar to the estimate for South Korea [39]. Dividing deaths-to-date by cases-to-date leads to a biased estimate of CFR, called naive CFR (nCFR) as the delays from confirmation of a case to death is not accounted for, as well as due to under-reporting of cases and even deaths. The reported numbers agree with recently published study for symptomatic case fatality ratio in China [40].

**Infection fatality, intensive care and hospitalisation ratios.** Infection fatality ratio (IFR) estimates are based on the study of [35], which reported IFR of 0.66% with 95% confidence interval 0.4% to 1.3%. These estimates are consistent with IFR estimate on Princess Diamond

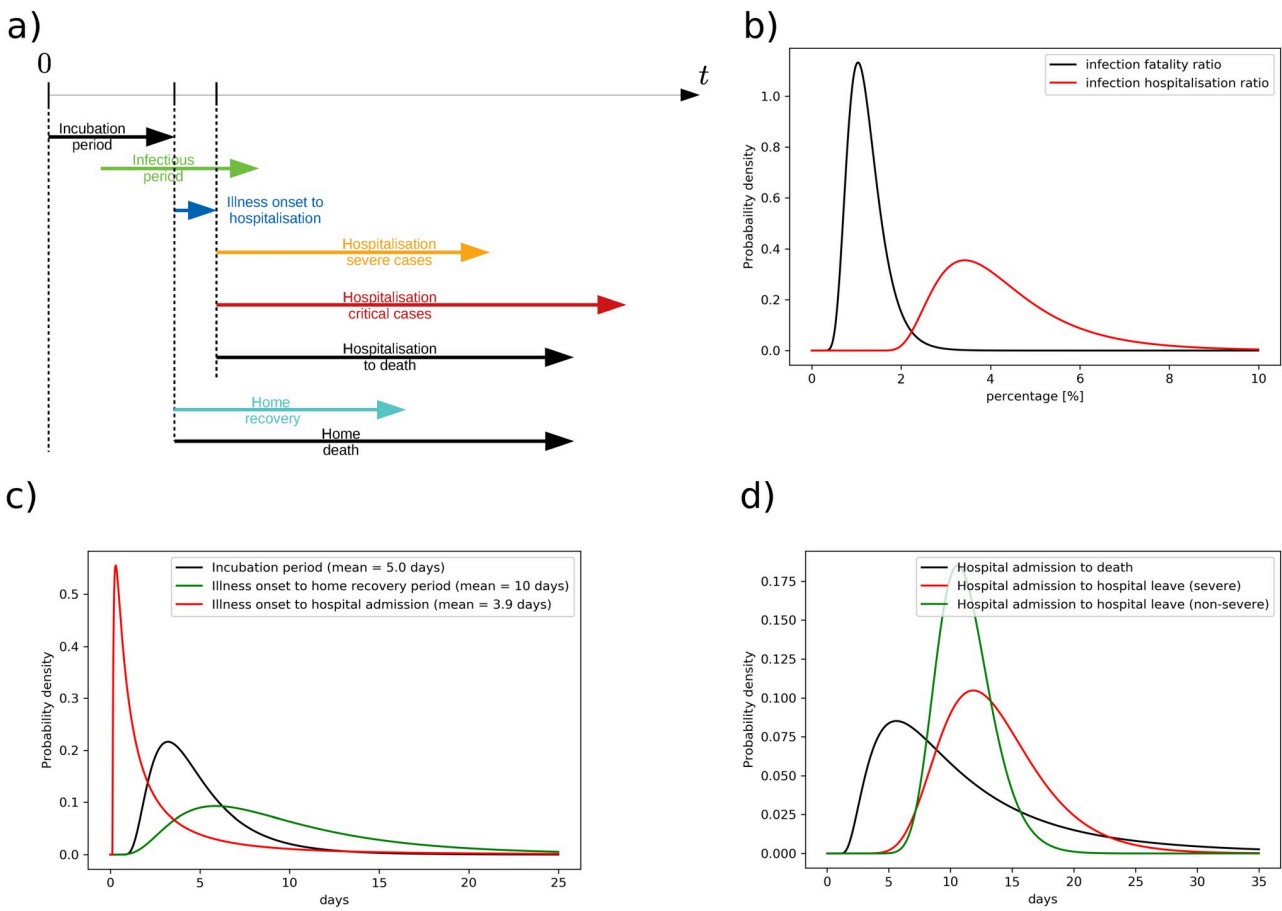

**Fig 4. Assumed COVID-19 illness evolution.** a) A simplified sketch of illness evolution. b) Infection fatality ratio distribution and infection hospitalisation ratio distribution for ensemble simulations. Computed based on data from [35]. c) Incubation period and illness onset to hospitalisation distribution for COVID-19 patients [36]. d) Mean distribution of hospital admission to death, hospital admission to hospital leave for severe and for non-severe illness [36, 37].

Cruise ship, when demographic differences are accounted for [41]. In Imperial College report on COVID-19 [42], these numbers have been also adjusted for the non-uniform attack rate and UK demography. The authors obtained age-stratified IFR estimates by adjusting their CFR estimates using COVID-19 prevalence data for expatriates evacuated from Wuhan. This approach involves very large uncertainties. Furthermore, [35] collected data from patients who were hospitalised in Hubei, mainland China, where median age is 37.4 years while median age in Slovenia is 44.5 years. Study reported a strong age gradient in risk of death. We have applied those age-stratified estimates to the Slovenian population. Performing an age-stratified weighted average, we compute the total IFR of 1.16% (95% CI 0.63-2.22%). Similar total IFR was reported by a comprehensive study for Italy (1.29%, 95% CI 0.89-2.01%) [43]. On the other hand somewhat lower IFR (0.95%, 95% CI 0.47-1.70%) has been estimated for Lombardia with the lower bound of 0.65%, consistent with 0.58% lower bound for Bergamo province [44]. A bit higher IFR of 1.6% (95% CI 1.1-2.1%) was reported in another study, while a thorough meta-analysis of IFR estimates was done by [45].

Analogously as for IFR, we compute the average hospitalisation rate of 6.37% (95% CI 3.8-13%) based on [35]. Slightly lower age-dependent hospitalisation rates were estimated for COVID-19 patients in USA [46], which adjusted for demography of Slovenia (but not

accounting for non-uniform attack rate) gives hospitalisation ratio of 3.97%. The latter result better coincides with the observed number of hospitalisations in Slovenia. No interval estimate is given, thus we use the same relative error as given by [35]. The final hospitalisation ratio is thus 3.97% (95% CI 2.37-8.10%). We assume that roughly one fourth to one third of all hospitalised cases are admitted to ICU [47], despite some studies showing smaller proportions [48]. We assume that one half of cases admitted to intensive care unit (ICU) are fatal [49].

Taking into account infection fatality ratio, hospitalisation ratio and ICU admission ratio, it follows that roughly one half of all deaths occur at home/elderly care center/palliative care center, which agrees with the present data for Slovenia [50]. Note that for simplicity, we have assumed uniform attack rate across all ages, despite studies showing that working population is most likely to get infected [51, 52]. Using the minimization procedures, we obtain parameters of log-normal distribution which best fits both values and their 95% confidence interval (Fig 4b).

**Incubation period—Infection to illness onset.**   Mean incubation period is taken to be 5 days (95% CI 4.2-6.0 days), while the 95th percentile of the distribution was 10.6 days (95% CI 8.5-14.1 days) and 99th percentile 15.4 days (99% CI 11.7-22.5 days) [36]. Similar numbers were reported in earlier studies with less patients included [24, 53–55]. Log-normal distribution is used to describe the distribution of incubation period among nodes. However, the parameters of the lognormal distribution also vary for every ensemble member, according to their uncertainty. Incubation period distribution and other outcome parameters are shown in Fig 4c.

**Infectious period.**   The infectious period is not yet well defined. A small study from German cohort of only 9 patients with mild clinical courses showed that viral shedding was high during the first week of symptoms and peaking at day 4 [56]. Another study from Singapore reported seven clusters in which virus was transmitted from a COVID-19 patient before experiencing symptoms. According to the authors pre-symptomatic transmission occurred 1-3 days before symptoms onset [57]. We have therefore estimated latent (non-infectious) period of 2.5 days and infectious period to start 2.5 days before the completion of incubation period (average incubation period is estimated at 5 days). Thus, we assume 2.5 days of pre-symptomatic transmission. Slightly larger numbers (2.55 days for Singapore and 2.89 days for Tianjin, China) were reported by [58].

The infectious period likely ends around 5 days from symptoms onset, so the total period of infectiousness lasts $T_{bioinf} \approx 7$ days. Note however that none of the interval boundaries are known exactly. Determining its final boundary is especially challenging, as it depends on the social factors as well, e.g. whether the infected cases are able to self-isolate from surroundings and how strictly they follow the self-isolation order. Here, we assume strict (100%) self-isolation and use a *social* infectious period of $T_{socinf} = 5$ days. It starts 2.5 days (95% CI 1.5-3.5 days) after infection and ends 2.5 days after incubation (95% CI 1.5-3.5 days) as the case ascertainment typically occurs 2 days after symptoms onset [59]. The infectious period fall in line with study of [34]. It also falls in line with the reported proportion of pre-symptomatic transmission (representing half of infectious period) being 48% for Singapore, 62% for Tianjin, China [60] and 44% for 77 infector-infectee pairs in Gaungzhou, China [61].

**Illness onset to hospitalisation or home recovery.**   From the illness onset on, there are two possible recovery pathways: home recovery/death or hospitalisation (Fig 4c). Home recovery period for mild cases has not been documented officially but is reported to be within one and two weeks. Since it does not affect the hospitalisation statistics, we here assume it to be log-normally distributed with mean period of 10 days.

Based on the clinical study of [36], mean illness onset to hospital admission period is 3.9 days (95% CI 2.9-5.3 days), with median of 1.5 days (95% CI 1.2-1.9 days), 5% percentile at 0.2

days (95% CI 0.1-0.3 days) and 95% percentile at 14 days (95% CI 10.3-20.1 days). Only the distribution of data for living patients is accounted for, since we now understand the severity of the illness.

**Hospital admission to recovery or death.** Hospital admission to death median (mean) length is assumed 6.7 (8.6) days long (Fig 4d). Only slightly longer periods were reported by [62] with mean length of 10.1 days. Hospital admission to recovery is on average longer than hospital admission to death. The median hospitalisation length is 11 days (95% CI 10-13) for non-severe cases and 13 days for severe (95% CI 11-17) [36]. Both are log-normally distributed. For ensemble computations, their medians are further log-normally distributed according to their respective confidence intervals. Similar numbers were reported by [63] with 11 day (95% CI 7-14) mean hospital length of stay and 8 day (95% CI 4-12) mean ICU length of stay.

Fatality ratio of severe cases in need of intensive care is reported to be around 50%. We assume fatality ratio of severe cases without intensive care to be normally distributed with mean of 90% (95% CI 85-95%). Fatality ratio of severely ill without oxygen is assumed to be 10% (95% CI 5-15%).

## Initial condition

The initial condition for the simulation is defined for March 12, 2020. To that day, there were 131 symptomatic cases who tested positive in Slovenia, 8 days after first positive case, which implies an anomalously low doubling time of $\tau = 1.23$ days. This number is case specific as there was winter holiday in Slovenia at the end of February and beginning of March. Thus, lots of cases were imported from Northern Italy (including Lombardy). Other studies typically suggest a doubling time of around 5 days (95% CI 4.3—6.2) in the initial uncontrolled stage of the epidemic [64]. Smaller values of around 3.5 days in most of Western Europe [65]. Thus, our choice is doubling time of $T_{double} = 3.5$ days (95% CI 2.5-4.5 days) for the period before March 12.

Different numbers of actually infected people were suggested in the media reports, ranging from 5 to 20 times the number of reported positive cases. Given the average incubation period of 5 days + (2 days for case ascertainment) and doubling period of 3.5 days, factor $2^{\frac{T_{inc}+2}{T_{double}}} = 4$ applies. Furthermore, the proportion of asymptomatic cases is around 18% based on the data from Diamond Princess Cruise Ship [66] (mostly older people) and around 33% based on the more recent study [67]. Population screening tests from Iceland reported 41.6% of all who tested positive, did not experience any COVID-19 symptoms [68]. Similar asymptomatic ratio of 43.2% (95% CI 32.2-54.7%) was reported also from a screening study conducted for the Italian town Vo [69]. Another study on the homeless population in Boston reported even larger proportion of asymptomatic cases [70]. The model-driven study of [71] found that 74% (95% CI 70-78%) of SARS-CoV-2 infections proceeded asymptomatically, raising also doubts about the assumed IFR. However, in this study, we opt for 40%, normally distributed with standard deviation of 10%. Furthermore, we double the value to account for the initial under-reporting of symptomatic cases, estimated by [22]. All together, this results in almost 1800 infected people in Slovenia by March 12.

Based on the exponential growth in the initial stage of the epidemic and known incubation period, we randomly generate the infection length of the patients with exponential distribution with shape factor of $T_{double}/\log2$, so that 131 develop symptoms and are ascertained by March 12. Initial distribution of 1780 infected people by the time-length of their infection is shown in Fig 5. Note that in reality, due to many imported cases, the actual infection-time distribution may be slightly different.

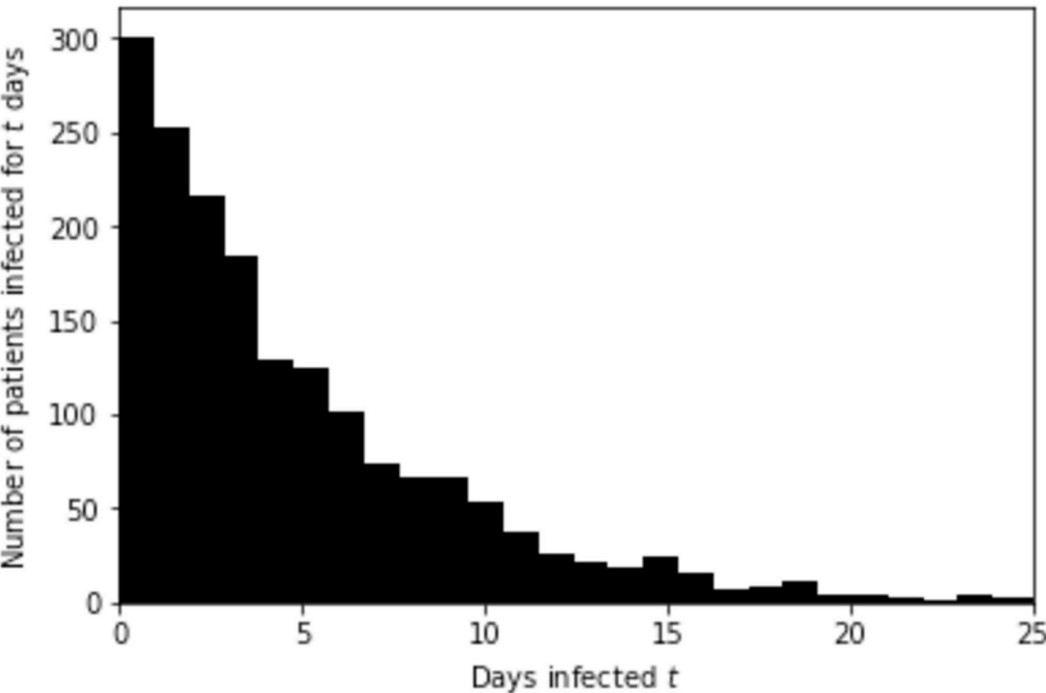

**Fig 5. Distribution of infected by the time since their infection.** Distribution of 1780 infected people on March 12, 2020, in Slovenia, by the time passed since their infection.

## Ensemble of simulations

Ensemble of simulations allows to estimate the uncertainty of the epidemic forecasts and to infer confidence in those predictions. There are two levels of perturbations in the ensemble: 1) at the start of each simulation, we perturb parameters, which govern the probability distributions of all model parameters. Thus, each simulation has slightly different probability distributions of its parameters. 2) each node in the network has its own transmission probability based on its number of contacts and each node has its own disease progression drawn from the associated probability distributions.

The uncertainty is associated with the impact of the intervention-measures on the social network connectivity and the uncertainty attributed to the intrinsic (internal, natural) model uncertainty. The latter can be further divided into:

1. social network uncertainty associated with randomized connections,

2. initial condition uncertainty as random nodes are infected,

3. virus transmission dynamics uncertainty which stems from the uncertainty of parameters, described in Virus transmission model,

4. disease progression model uncertainty due to the uncertainty of parameters, described in Disease progression model.

In Slovenia, the intervention measures were imposed at several time instances between March 13 and March 30 [72]. Their impact is assessed as follows. First, we perturb the social network connectivity (by perturbing the scale parameter $\theta$ in the Gamma distribution of the number of contacts, Eq 1) at the time instances, when intervention measures took place. Then we measure the discrepancy of each simulated run from the observed evolution of the number

of hospitalised patients ($H$), patients in intensive care unit ($ICU$) and fatal cases ($F$), using the cost function

$$J = \sum_{i=0}^{N} (|\log y_{ICU}(t_i) - \log x_{ICU}(t_i)| + |\log y_F(t_i) - \log x_F(t_i)| + |\log y_H(t_i) - \log x_H(t_i)|). \quad (6)$$

Logarithms are used to weigh equally the initial and later phase of the pandemic, as the number of infected varies by several orders of magnitude.

The final probabilistic forecast only consists of those ensemble members, for which the cost function $J$ is minimal. In practice, an ensemble of 1000 perturbed simulations was computed. Among all simulations, only 10% of simulations with smallest $J$ is used to generate the final probabilistic forecast. The described data assimilation approach allow us to estimate both the impact of the intervention measures as well as the changes in the distribution of parameters.

## Exclusion experiments

We perform exclusion experiments to assess the contribution of the above-mentioned model components uncertainty to the total forecast uncertainty. For example, to estimate the contribution of the randomized social network to the total forecast uncertainty, we run an ensemble of simulations with the same social network, i.e. we exclude the social network perturbation.

The proxy for forecast uncertainty is the relative spread, i.e. the spread of the forecast ensemble, divided by the median value of the forecasts at each time instance. As the spread is approximately symmetric on the logarithmic axis for phenomena with exponential dynamics [73], we compute the relative spread as:

$$RS(t) = \frac{\log P_{75}(\vec{x}(t)) - \log P_{25}(\vec{x}(t))}{\log \text{median}(\vec{x}(t))}, \quad (7)$$

where $P_{75}$ and $P_{25}$ indicate 75th and 25th percentiles of population $\vec{x}$ at time $t$.

## Results

### Prediction for Slovenia issued on May 5, 2020

Every day, new data is used to correct the COVID-19 forecast. Fig 6 shows an example of the ensemble prediction issued on May 5, 2020, simulated from the initial condition on March 12, 2020. The shown forecast is issued in the already declining stage of the epidemic and assumes ongoing intervention measures. Fig 6a shows 100 members (out of 1000), whose evolution least deviates from the observed data. Fig 6b shows the associated probabilistic forecast. The infectious population has the largest uncertainty relative to its value, however the number of infectious is not constrained by any measurements. Thus, its relative uncertainty roughly reflects the uncertainty in the hospitalisation, ICU and IFR ratios. The total number of infected to date approaches 11000 people (90% CI 7000-17000), in line with the estimate of the underreporting of symptomatic cases (only 17% of cases reported) at the time [41] and the estimated asymptomatic ratio of coronavirus infections [69].

In April 2020, a National COVID-19 prevalence survey has been completed, which reported 2 actively infected out of 1367 tested (prevalence 0.15%, 95% CI 0.03–0.47%) [74] and 41 positive for coronavirus antibodies out of 1318 tested (3.1% prevalence, 95% CI 2.2-4%) [75]. However at the time, the survey added little extra information to better constrain the forecast. First, the number of actively infected is associated with large confidence interval, and second, the antibody tests have significant false-positive rate and varying sensitivity [76]. Accounting for

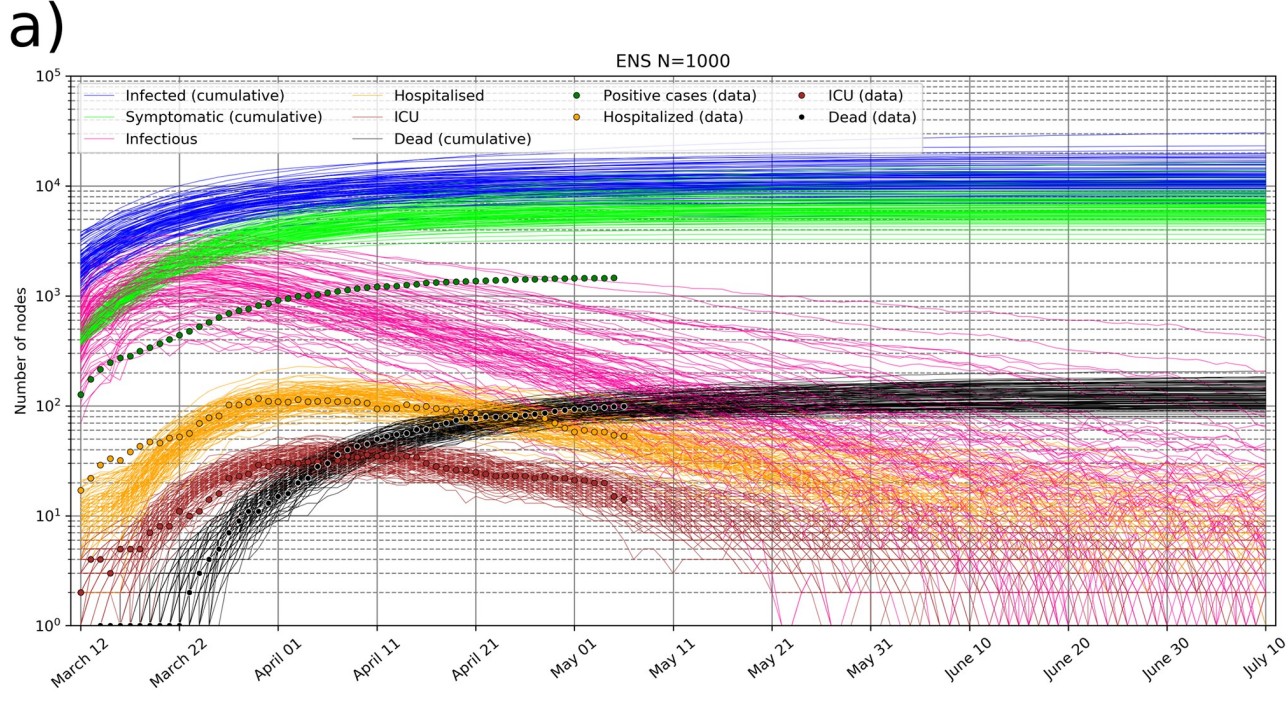

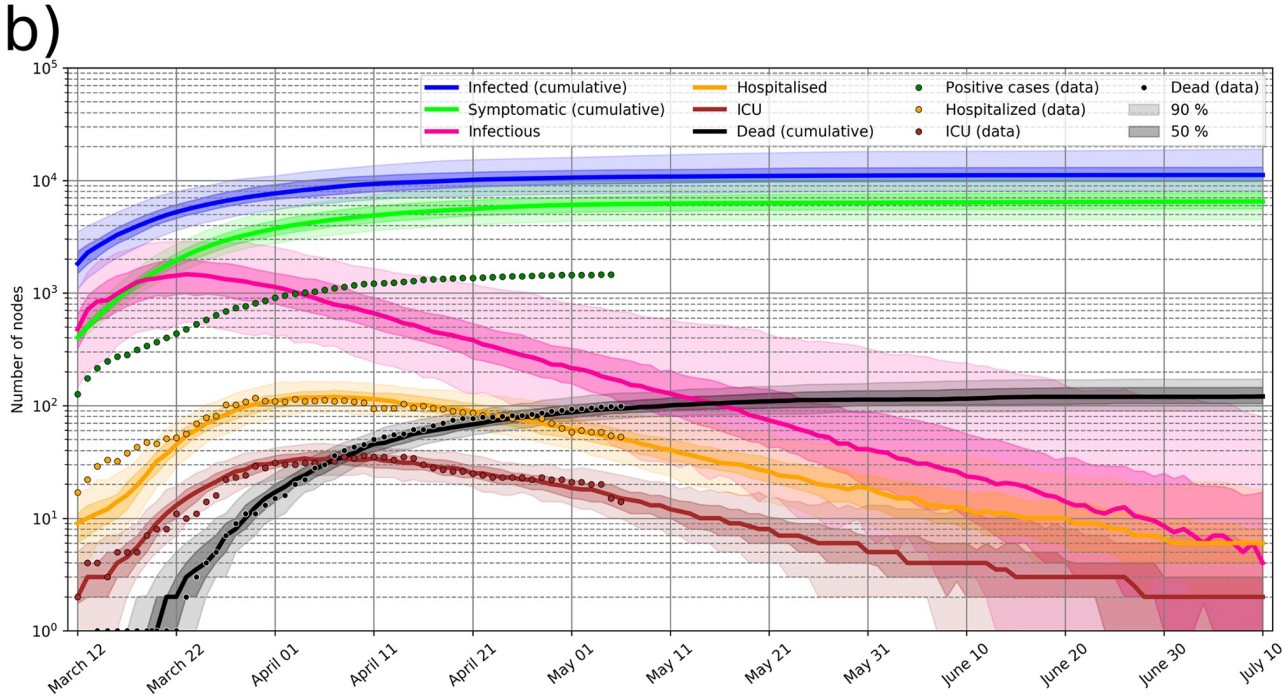

**Fig 6. Forecast of COVID-19 epidemic in Slovenia.** Forecast of COVID-19 epidemic in Slovenia issued on May 5, 2020 and comparison with real data. a) 100 ensemble members which best fit the observed data (dots) are shown. b) Probabilistic forecast: median value, interquartile range (50%; 25th-75th percentile) and 90% range are shown.

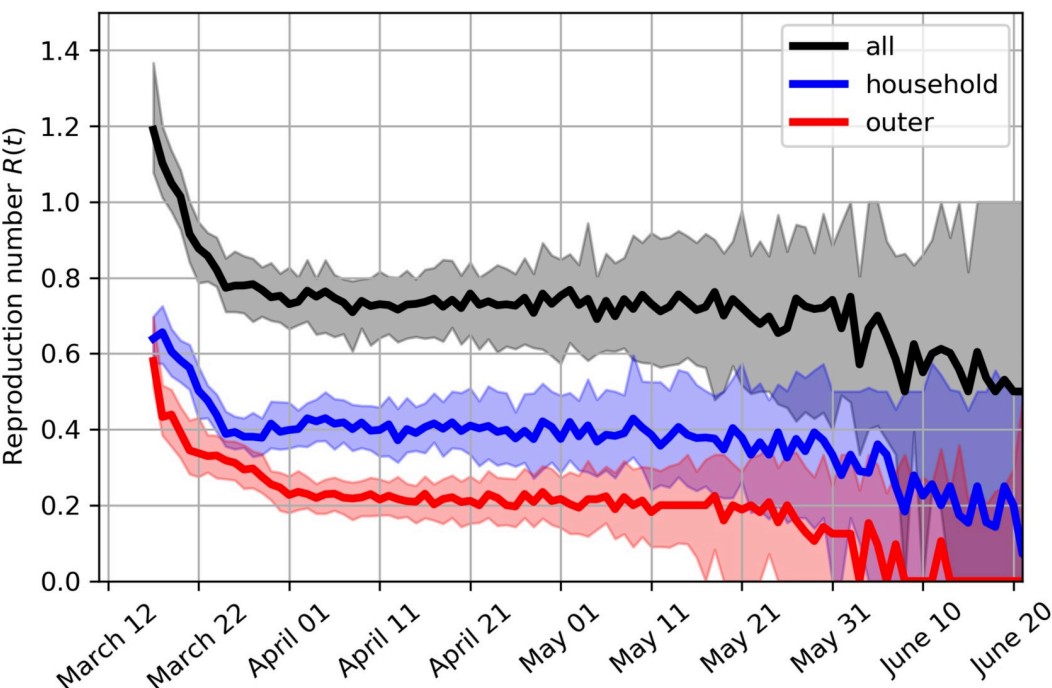

**Fig 7. Effective reproduction number.** Evolution of the estimated effective reproduction number $R(t)$, decomposed into reproduction number associated with household transmission and transmission outside households. The shaded regions indicate the interquartile ranges.

the latter, the posterior estimate of the SARS-CoV-2 seroprevalence was later estimated at 0.8% (95% CI 0-2.8%) [77].

In the social network model, the current reproduction number $R$ can be directly measured. For each infectious node, we count the number of nodes it infects. Then we assign the counts to the time instance corresponding to the end of the infectious period. Fig 7 shows the reproduction number falling below 1 on March 20, 2020, which marks the transition into decaying stage of the epidemic. Current estimate of $R$ is at around 0.75, in line with the recent estimate for Slovenia of [72]. Fig 7 also shows that the infection is currently much more likely to transmit within households than outside households. If the current intervention measures continue, the reproduction number would start to rapidly decline at the end of May without any extra intervention measures, which indicates effective virus containment when the virus would be transmitted only within some of the household clusters.

The members of the ensemble, which minimize the cost function, can also be used to inverse estimate the posterior distribution of clinical parameters, such as hospitalisation ratio, ICU ratio, ratio of severe infection, and IFR, as well as disease progress parameters such as the probability distribution of the time-span of hospital admission to death. For example, according to Fig 8a, the true hospitalisation rate is slightly smaller than the first guess, while the infection fatality rate is 0.1% higher in the posterior analysis. As another example, Fig 8b shows that the posterior estimate of the mean hospital admission to death duration is 7.5 days, half a day longer than the first guess estimate. This inverse technique was also used to estimate the impact of intervention measures on the social network connectivity. However, at the time, virus transmission parameters and some disease progress parameters (e.g. IFR) could not be constrained due to the lack of reliable data on the infectious population and total infected population.

a)

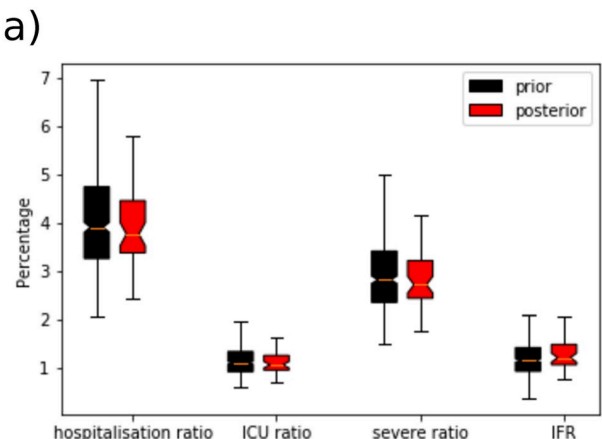

b)

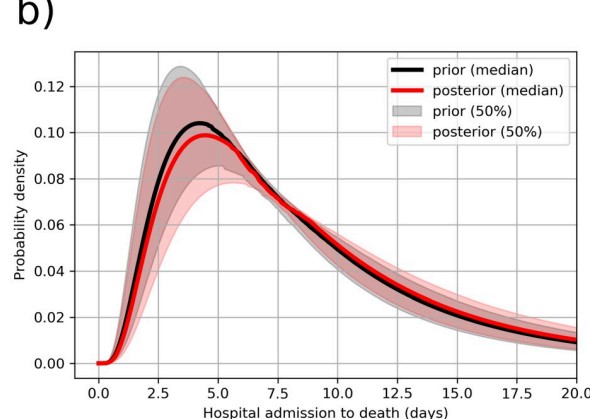

**Fig 8. Prior/posterior distributions of disease outcome ratios.** a) Prior and posterior distributions of hospitalisation ratio, intensive care unit (ICU) ratio, ratio of severe symptoms (requiring hospitalisation), and infection fatality ratio (IFR). b) The probability distribution of the duration of hospital admission to death.

## Forecast uncertainty decomposition

Using the exclusion experiments, we evaluated the contribution of different epidemic model components to the total forecast uncertainty of the total number of infected and infectious population. For instance, the ensemble experiment where the social network and the initial condition are fixed (not perturbed) is termed NONET, the experiment without virus transmission dynamics perturbation is called NOTRANS, while the experiment without disease progression model perturbation is named NODIS.

We perform exclusion experiments for two different cases: uncontrolled epidemic and controlled epidemic with intervention measures and low infected population. The results are shown in Fig 9. We observe, that in the uncontrolled epidemic, the forecast uncertainty is most reduced when the transmission dynamic parameters are not perturbed (experiment NOTRANS in Fig 9a and 9b). This also reduces the uncertainty in the epidemic peak and later stages of the epidemic. Fixing disease progression parameters (such as ratio of asymptomatic infections and duration of infectiousness) also significantly reduces uncertainty (experiment NODIS). Fixing initial condition and social network structure reduces the uncertainty only in the initial stage of the epidemic (until around day 10), when the number of infected individuals is small (experiment NONET) and homogeneous mixing is an invalid assumption. In the later stage, the uncertainty becomes similar to the basic experiment with all parameters perturbed (experiment ALL). These experiments indicate that the largest contributor to the forecast uncertainty in the uncontrolled epidemic is virus transmission dynamics.

In the controlled epidemic with low number of infected, though, fixing the social network and initial condition (NONET, Fig 9c) reduces the forecast uncertainty the most (the impact is amplified again in the initial stage), followed by fixing the disease progression parameters, with the impact amplified again in the early part of the simulation. This suggests that the structure of the network and the initial distribution of infected nodes drastically affects the evolution due to heterogeneous mixing and randomized irregular social network. The result suggests that the epidemic forecast can be improved (i.e. its uncertainty decreased) the most by constructing a more realistic model of our social network.

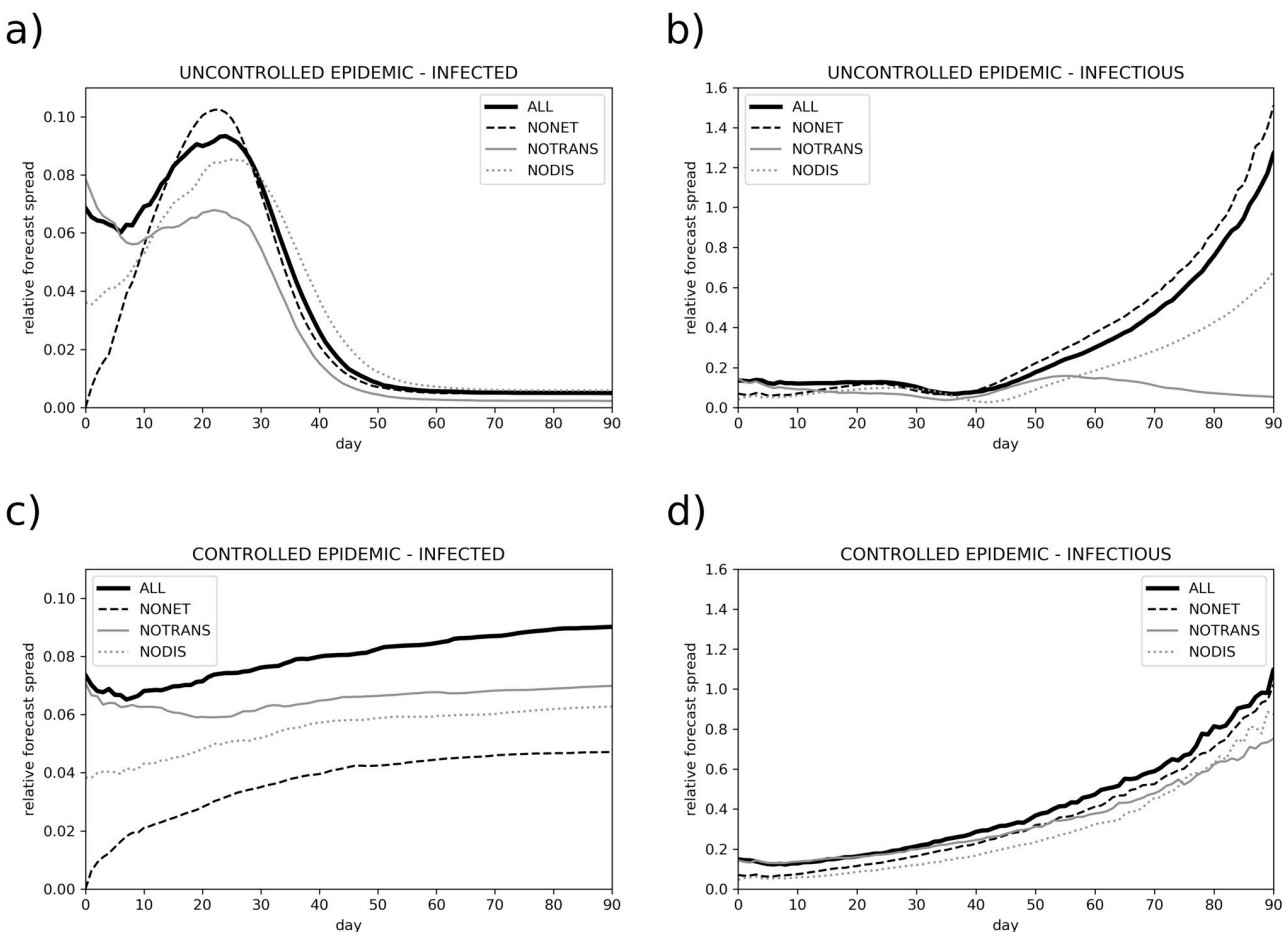

**Fig 9. Relative forecast spread.** Relative forecast spread, measured by Eq 7, for uncontrolled epidemic (a,b) and controlled epidemic with low number of infected (c,d), as shown in Fig 6. The basic experiment with all parameters perturbed is termed ALL. NONET stands for no social network and initial condition perturbation, NOTRANS stands for no transmission dynamics parameters perturbations, while NODIS means no disease progress model parameters perturbations.

## Discussion, conclusions and further outlook

In this study, we have developed a virus transmission model on the simplified social network of Slovenia with 2 million nodes organised into home/care center clusters. A detailed disease progression model is coupled with the virus transmission model. The model probabilistic prediction has been regularly updated on Sledilnik webpage [50] since the end of March and is occasionally communicated to the Expert Group that provides support to the Government of the Republic of Slovenia for the containment and control of the COVID-19. The model software is available in S1 File.

We have developed a simple data assimilation procedure, which minimizes the cost function measuring the deviation from the observed ICU, hospitalisation and fatality values. The procedure constrains the forecast trajectories closer to the observed values, while it also constrains the model parameters. Our approach somewhat mimics the established variational data assimilation (DA) approach in Numerical Weather Prediction (NWP) [78, 79]. Several others have utilised variational DA in epidemiology [80], ensemble DA [81] with Ensemble Adjustment Kalman Filter [82]. Most recently, a comprehensive assessment of the COVID-19

pandemic was performed using iterative ensemble smoother [83]—the ensemble smoother with multiple data assimilation [84].

An indispensable part of the prediction is its uncertainty. In this study, we evaluated the contribution of the virus transmission uncertainty (e.g. reproduction number and its derivatives), network and initial condition uncertainty and uncertainty of the disease progress model to the total uncertainty of the epidemic forecast. We found that in the uncontrolled epidemic, the intrinsic uncertainty mostly originates from the uncertainty of the virus transmission, while the randomness of the social network has only minor impact of the final size of the epidemic. The latter is in line with a study, where the social network was constructed based on extensive contact survey data, and which reported only minor impact of reshaping the network structure or removing the variance of connection weights on the final size of the epidemic. On the opposite, in the controlled epidemic with low infected population, the randomness of the social network becomes the major source of forecast uncertainty. We also show, that the uncertainty of the forecast and the associated risk is extremely asymmetric (roughly symmetric on a logarithmic axis) with long exponential tails, reaching a similar conclusion to the recent study of [73].

There are some limitations of our model which reduce its predictive ability and its usefulness to simulate the impact of intervention measures in advance. Our social network model is too simplified: the connections among nodes outside households are quasi-static in time, but have no realistic topological structure. Thus, the average clustering in our network model is most likely too low, as in the real-world social networks people typically interact within densely connected social groups [11]. In the real-world social networks, some connections are more risky than other, while our model does not account for that by e.g. weighting the network connections. Furthermore, regional work/education clustering based on work/education mobility data is not included in the present social network. The nodes do not have attributions such as age, sex or employment status and the social mixing data [18, 52, 85, 86] is not accounted for yet. Given the high attack rate within households, the social mixing within households is of special importance, thus it is also vital to include the age-distribution of the residents of different household sizes. A more sophisticated treatment of the secondary attack rate is also needed, for example the infectiousness could be modeled as a function of time [61, 87]. Further work should alleviate some of the mentioned limitations to allow more robust simulation of the intervention measures.

The ongoing COVID-19 epidemic has revealed a major gap in our ability to forecast the evolution of the epidemic. No operational center for infectious disease prediction, similar to those employed for the weather predictions (e.g. European Centre for Medium-Range Forecasts or National Center for Environmental Prediction), exists, despite the gigantic societal, economical and health impact of the ongoing epidemic. While the epidemic dynamics is governed by the human social behaviour and its modeling is arguably messier than weather forecasting [88], a coordinated modeling effort which borrows the established methods used for Numerical Weather Prediction (NWP) would likely improve our prediction [81]. Accurate models of the real-world social networks are needed to realistically simulate the virus transmission dynamics. Similarly to NWP models [89], the real-time clinical patient data, mobility data [90] and connectivity data (obtained by e.g. postprocessing the bluetooth-generated anonymous contact data [91]), should be rapidly assimilated into the virus spread prognostic model [92] to evaluate the changes in contact patterns [93]. This would allow 1) to estimate the critical virus spread parameters and their uncertainty, 2) to forecast the evolution of the epidemic more accurately and based on that forecasts, 3) to implement optimal worldwide-concerted measures to minimize the virus spread. We should be ready for the next big pandemic!.

## Supporting information

**S1 File. Model software.** The core program `korona_final.py` is written in Python 2.7 and requires standard `scipy`, `numpy` and `matplotlib`. The most time-consuming procedures of the software are written in Fortran 90. Python binding are created using F2PY [94]:
`f2py -c generate_connections.f90 -m generate_connections`.
(ZIP)

## Acknowledgments

The authors are grateful to Aleks Jakulin, Miha Kadunc (Sinergise), Luka Renko (founder of the volunteer-led Sledilnik.org project) for providing the timely COVID-19 pandemics data for Slovenia. Žiga Zaplotnik would like to thank Prof. Alojz Kodre and Asst. Prof. Simon Čopar (both University of Ljubljana, Faculty of Mathematics and Physics, UL-FMF) for introducing him the node-based analysis of the information spread. We thank fellow physicists Nejc Davidovič and Jan Bohinec (Gen-I) for fruitful discusions as well as Asst. Prof. Samo Drobne (University of Ljubljana, Faculty of civil and geodetic engineering) for providing the mobility data for Slovenia. Special thanks go to Prof. Roman Jerala (National Institute of Chemistry) and Prof. Tomaž Zwitter (UL-FMF) for discussions of the model, and for reading and commenting the early manuscript draft as well as for providing useful literature. Finally, Žiga Zaplotnik would like to thank his postdoc supervisor Prof. Nedjeljka Žagar (University of Hamburg) for always supporting work beneficial to society.

## Author Contributions

**Data curation:** Aleksandar Gavrić.

**Formal analysis:** Žiga Zaplotnik.

**Investigation:** Žiga Zaplotnik, Aleksandar Gavrić, Luka Medic.

**Methodology:** Žiga Zaplotnik, Luka Medic.

**Software:** Žiga Zaplotnik, Luka Medic.

**Supervision:** Žiga Zaplotnik.

**Validation:** Aleksandar Gavrić.

**Visualization:** Žiga Zaplotnik.

**Writing – original draft:** Žiga Zaplotnik, Aleksandar Gavrić.

**Writing – review & editing:** Žiga Zaplotnik, Aleksandar Gavrić, Luka Medic.

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
