## [Decision Letter · Decision Letter 0]

3 Jul 2020

PONE-D-20-15323

Simulation of the COVID-19 epidemic on the social network of Slovenia: estimating the intrinsic forecast uncertainty

PLOS ONE

Dear Dr. Zaplotnik,

Thank you for submitting your manuscript to PLOS ONE. After careful consideration, we feel that it has merit but does not fully meet PLOS ONE’s publication criteria as it currently stands. Therefore, we invite you to submit a revised version of the manuscript that addresses the points raised during the review process.

Please respond to the reviewer comments on a point-by-point basis and revise the manuscript accordingly.

We look forward to receiving your revised manuscript.

Kind regards,

Jeffrey Shaman

Academic Editor

PLOS ONE

Journal Requirements:

Reviewers' comments:

Reviewer's Responses to Questions

**Comments to the Author**

1. Is the manuscript technically sound, and do the data support the conclusions?

Reviewer #1: Yes

Reviewer #2: Yes

2. Has the statistical analysis been performed appropriately and rigorously? 

Reviewer #1: Yes

Reviewer #2: Yes

3. Have the authors made all data underlying the findings in their manuscript fully available?

Reviewer #1: Yes

Reviewer #2: Yes

4. Is the manuscript presented in an intelligible fashion and written in standard English?

Reviewer #1: Yes

Reviewer #2: Yes

5. Review Comments to the Author

Reviewer #1: Comment on: PONE-D-20-15323

This paper discussed the spread of the COVID-19 epidemic due to the contact between household person and out it using the social network model of Slovenia. The paper is well-organized and the other parts the paper is well-presented. Authors formulate the paper in a very traditional way and its very suitable for non-mathematical readers, especially important in this topic. However, there are some major issues with the paper.

First of all, In the fourth line of the introduction, the authors refer to the variable I as an immunity and this is a wrong while it represents the infected as indicated in the paper.

Second, is there any relevant Authors did not mention or provide any related concept or references to illustrate the reason of their approach. So, must for authors write some references and show what is different from this study.

Last, I am advised to accepted to published this paper.

Reviewer #2: Manuscript Number: PONE-D-20-15323

Simulation of the COVID-19 epidemic on the social network of Slovenia: estimating the intrinsic forecast uncertainty

_Ziga Zaplotnik1*, Aleksandar Gavri_c2, Luka Medic1

I find the “ Simulation of the COVID-19 epidemic on the social network of Slovenia” to be a good contribution to the study of possible avenues for spread and possible social controls of COVID-19 at country level! To achieve this, the authors have cited and utilized similar situations in other countries. Countries where such formulations and analyses have not yet been employed, can easily adopt the Slovenia social network model presented and simulated in the manuscript.

Where I have found it necessary for the authors to clarify/justify, I have indicated so as in the following table and/or I have annotated some of my comments in the manuscript.

Page Line(s) Comment

1-2 14 - 16 The idea of “superspreaders” is very important.

31 - 36 Sections 2, 3, 4 are referred to in the paragraph, but other than the change in size of the font of the headings of the sections/subsections, the sections have no numerical labels in the manuscript. This should be rectified.

Table 1 Shouldn't the caption be at the top of the table; or is it the style of the journal?

3 Eqn. (1) It would be good to remind the reader what k and θ stand for!

4 100 - 101 Each of “hospitalised with severe illness, hospitalised and critically ill (ICU treatment), hospitalised with fatal outcome” should have been denoted and considered differently!

7 209 labels a) and b) are missing from Fig 4.

8 252 - 253 “fatal cases being admitted to hospital” not clear to the reader!

12 - 13 440 - 451 These are good recommendations, but definitely more involving!

6. PLOS authors have the option to publish the peer review history of their article (what does this mean?). If published, this will include your full peer review and any attached files.

Reviewer #1: No

Reviewer #2: **Yes: **Livingstone Luboobi

---

## [Author Response · Author response to Decision Letter 0]

30 Jul 2020

Response to the Reviewers’ comments on PONE-D-20-15323 manuscript

Simulation of the COVID-19 epidemic on the social network

of Slovenia: estimating the intrinsic forecast uncertainty

by Žiga Zaplotnik, Aleksandar Gavrić and Luka Medic

Dear Prof. Shaman,

thank you very much for the reviews of our paper.

In response to Reviewers’ comments, we have performed a revision of our paper. We have corrected citations (checked whether the cited preprints have been published) and described supporting information (model software). Following Reviewer #1’s request, we performed a review of similar network and data assimilation approaches in epidemiology and emphasised more the reasons for our computational approach. Thanks to Reviewer #2’s detailed review, we corrected errors in the manuscript and also further improved the text for better clarity.

We hope that the revised paper is now better suited for publication in PLOS ONE.

Enclosed please find our point-by-point responses to the Reviewers’ comments using the same organisation as in their review. Comments are coloured blue whereas our responses are in black. 

Your sincerely,

Žiga Zaplotnik, Aleksandar Gavrić, Luka Medic

Response to the comments of Reviewer #1

General comments:

This paper discussed the spread of the COVID-19 epidemic due to the contact between household person and out it using the social network model of Slovenia. The paper is well-organized and the other parts the paper is well-presented. Authors formulate the paper in a very traditional way and its very suitable for non-mathematical readers, especially important in this topic. However, there are some major issues with the paper.

First of all, In the fourth line of the introduction, the authors refer to the variable I as an immunity and this is a wrong while it represents the infected as indicated in the paper.

Second, is there any relevant Authors did not mention or provide any related concept or references to illustrate the reason of their approach. So, must for authors write some references and show what is different from this study.

Last, I am advised to accepted to published this paper.

Response:

We thank the Reviewer for her/his review and constructive comments. 

We corrected the blunder in line 4. Letter “I” stands for infected population and not immune. Thank you for noticing this error.

We added a short review of previous approaches with network models in epidemiology and emphasised the reasons for network-based approach in our study. The text now reads:

“In order to account for the superspreading nature of the new coronavirus

(SARS-CoV-2) and to properly estimate the forecast uncertainty, we use network-based

approach to simulate the virus spread. The simplified social network consists of more

than 2 million nodes with a total of up to 20 million undirected connections,

representing the population of Slovenia and the contacts of its inhabitants, with realistic

distinction between household and outer contacts. Despite being computationally more

expensive, the advantage of the network approach is that it allows direct simulation of

intervention measures, contact-tracing strategies and other strategies of the the virus

containment [7, 8] as well as the lockdown-exit strategies.

The network epidemiology research has heavily advanced in the last three

decades [8]. A variety of different network types has been developed [9], however the

breakthrough of social-network approaches has been halted by the insufficient social

data and epidemiological data which would allow to verify different assumptions in the

generation of social networks [10]. An exception to this includes studies, where the

social network was generated based on the comprehensive contact survey data [11, 12].

Nevertheless, network models have often been criticised for the large number of

parameters they require [13].

In this study, we perform an ensemble-of-simulations of the virus spread over the

social network. Since the network is randomly generated in each simulation, the

evolution of the epidemics will differ between simulations. Different nodes are infected

at initial time, while each simulation also uses different virus transmission parameters

and disease progress parameters, which are perturbed according to their known

distributions. This approach allows to estimate the uncertainty of the epidemic forecasts

in the case of controlled epidemic and uncontrolled epidemic. To our knowledge, no

study has ever compared the impact of network perturbations, transmission parameters

perturbations and other perturbations on the uncertainty of the epidemic forecast.”

Response to the comments of Reviewer #2

General comments:

I find the “ Simulation of the COVID-19 epidemic on the social network of Slovenia” to be a good contribution to the study of possible avenues for spread and possible social controls of COVID-19 at country level! To achieve this, the authors have cited and utilized similar situations in other countries. Countries where such formulations and analyses have not yet been employed, can easily adopt the Slovenia social network model presented and simulated in the manuscript.

Where I have found it necessary for the authors to clarify/justify, I have indicated so as in the following table and/or I have annotated some of my comments in the manuscript.

Response:

We thank Prof.Livingstone Luboobi for his detailed review and annotations. Based on the fast growth of global computational resources, we believe that the future epidemiological models must resolve every individual and incorporate as much data as possible. Slovenia is due to its small population a perfect example for such case study.

Page Line(s) Comment

1-2 14 - 16 The idea of “superspreaders” is very important.

Response: we also believe so. There have been further evidences of superspreading nature of SARS-CoV-2 that we have included in the citations.

31 - 36 Sections 2, 3, 4 are referred to in the paragraph, but other than the change in size of the font of the headings of the sections/subsections, the sections have no numerical labels in the manuscript. This should be rectified.

Response: corrected as suggested. The text now reads: 

“Methodology section describes the social network model, the virus transmission model and the coupled disease progression model. The probabilistic ensemble forecast of the COVID-19 epidemic for Slovenia and the contribution of different model components to the total forecast uncertainty are described in section Results, followed by the discussion, conlusions and further outlook on the modelling of SARS-CoV-2 virus spread.”

Table 1 Shouldn't the caption be at the top of the table; or is it the style of the journal?

Response: according to the formatting sample, it should be at the top. Corrected!

Eqn. (1) It would be good to remind the reader what k and θ stand for!

Response: we have modified the text as follows: “In this study, we use Gamma distribution with shape parameter k = 1.65 and scale parameter θ = 4.08 for the initial setup in order to mimic the above mentioned negative binomial distribution.”

100 - 101 Each of “hospitalised with severe illness, hospitalised and critically ill (ICU treatment), hospitalised with fatal outcome” should have been denoted and considered differently!

Response: the text is removed in the “Compartments” subsection for clarity, since this division is part of the disease progression model.

209 labels a) and b) are missing from Fig 4.

Response: corrected!

252 - 253 “fatal cases being admitted to hospital” not clear to the reader!

Response: while the sentence was unclear, it also did not have direct impact on the model, thus it was removed.

440 - 451 These are good recommendations, but definitely more involving!

Response: We agree, implementing all of these would require lots of effort and most likely a larger team! Meanwhile, we have already implemented the contact-tracing strategies, mask usage, as well as mobility data.

---

## [Decision Letter · Decision Letter 1]

11 Aug 2020

Simulation of the COVID-19 epidemic on the social network of Slovenia: estimating the intrinsic forecast uncertainty

PONE-D-20-15323R1

Dear Dr. Zaplotnik,

We’re pleased to inform you that your manuscript has been judged scientifically suitable for publication and will be formally accepted for publication once it meets all outstanding technical requirements.

Kind regards,

Jeffrey Shaman

Academic Editor

PLOS ONE

Additional Editor Comments (optional):

Reviewers' comments:

Reviewer's Responses to Questions

**Comments to the Author**

1. If the authors have adequately addressed your comments raised in a previous round of review and you feel that this manuscript is now acceptable for publication, you may indicate that here to bypass the “Comments to the Author” section, enter your conflict of interest statement in the “Confidential to Editor” section, and submit your "Accept" recommendation.

Reviewer #1: All comments have been addressed

Reviewer #2: All comments have been addressed

2. Is the manuscript technically sound, and do the data support the conclusions?

Reviewer #1: Yes

Reviewer #2: Yes

3. Has the statistical analysis been performed appropriately and rigorously? 

Reviewer #1: Yes

Reviewer #2: Yes

4. Have the authors made all data underlying the findings in their manuscript fully available?

Reviewer #1: Yes

Reviewer #2: Yes

5. Is the manuscript presented in an intelligible fashion and written in standard English?

Reviewer #1: Yes

Reviewer #2: Yes

6. Review Comments to the Author

Reviewer #1: (No Response)

Reviewer #2: I am happy that you have responded positively to my comments on the first edition of the manuscript. For example, reference to the different sections of the manuscript, is now clear.

7. PLOS authors have the option to publish the peer review history of their article (what does this mean?). If published, this will include your full peer review and any attached files.

Reviewer #1: **Yes: **Ahmed Ali Mohsen

Reviewer #2: **Yes: **Livingstone Serwadda Luboobi

---

## [Editor Report · Acceptance letter]

17 Aug 2020

PONE-D-20-15323R1 

Simulation of the COVID-19 epidemic on the social network of Slovenia: estimating the intrinsic forecast uncertainty 

Dear Dr. Zaplotnik:

I'm pleased to inform you that your manuscript has been deemed suitable for publication in PLOS ONE. Congratulations! Your manuscript is now with our production department. 

Kind regards, 

on behalf of

Prof. Jeffrey Shaman 

Academic Editor

PLOS ONE